# From Hamamelitannin Synthesis to the Study of Enzymatic Acylations of D-Hamamelose

**DOI:** 10.3390/biom13030519

**Published:** 2023-03-12

**Authors:** Mária Mastihubová, Vladimír Mastihuba

**Affiliations:** Institute of Chemistry, Slovak Academy of Sciences, Dúbravská Cesta 9, 845 38 Bratislava, Slovakia

**Keywords:** hamamelitannin, D-hamamelose, regioselective acylation, hydrophobized gallates, enzymatic galloylation, Lipozyme TL IM

## Abstract

The bioactive natural substance, hamamelitannin, was effectively synthesized in two ways. The chemical acylation of 2,3-*O*-isopropylidene-α,β-D-hamamelofuranose promoted by Bu_2_SnO using 3,4,5-tri-*O*-acetylgalloyl chloride, followed by the deprotection provided hamamelitannin in 79%. Pilot enzymatic benzoylation of D-hamamelose using vinyl benzoate (4 equiv.) and Lipozyme TL IM as a biocatalyst in *t*-butyl methyl ether (*t*-BuMeO) gave mainly benzoylated furanoses (89%), of which tribenzoates reached (52%). Enzymatic galloylation of 2,3-*O*-isopropylidene-α,β-D-hamamelofuranose with vinyl gallate under the catalysis of Lipozyme TL IM in *t*-butyl alcohol (*t*-BuOH) or *t*-BuMeO provided only the 5-*O*-galloylated product. The reaction in *t*-BuMeO proceeded in a shorter reaction time (61 h) and higher yield (82%). The more hydrophobic vinyl 3,4,5-tri-*O*-acetylgallate in the same reactions gave large amounts of acetylated products. Vinyl gallate and triacetylgallate in the enzymatic acylation of D-hamamelose with Lipozyme TL IM in *t*-BuMeO yielded 2′,5-diacylated hamamelofuranoses in a yield below 20%. The use of other vinyl gallates hydrophobized by methylation or benzylation provided 2′,5-diacylated hamamelofuranoses in good yields (65–84%). The reaction with silylated vinyl gallate did not proceed. The best results were obtained with vinyl 2,3,5-tri-*O*-benzyl gallate, and the only product, 2′,5-diacylated hamamelofuranoside precipitated from the reaction mixture (84% in 96 h). After debenzylation, hamamelitannin was obtained an 82% yield from hamamelose in two steps. This synthesis is preparatively undemanding and opens the way to multigram preparations of bioactive hamamelitannin and its analogues.

## 1. Introduction

Tannins are polyphenolic plant secondary metabolites that have enormous structural diversity [1]. Hydrolyzable tannins are often isolated from plants for their remarkable therapeutic effects [2]. Their structure generally consists of a central sugar core, typically a glucose unit, to which galloyl groups, often *meta*-depsidically bonded (gallotannins) or C-C bonded dehydrodigalloyl units (ellagitannins), are attached [3]. The total synthesis of such compounds tends to be complex [4,5,6].

Hamamelitannin (2′,5-di-*O*-galloyl-2-*C*-(hydroxymethyl)-D-ribofuranose or 2,5-di-*O*-galloyl-D-hamamelofuranose, **1**, Figure 1) is the main component of the witch hazel bark extract (*Hamamelis virginiana* L.) [7,8,9]. Different galloylhamameloses, have been isolated to date from the bark of various Fagaceae spp. such as *Castanea crenata* [10,11], *Castanopsis cuspidate* [12], or *Castanea sativa* [13]. Gallotannin **1** is commercially available pure or as a component of various organic extracts. For commercial use, it is usually isolated from witch hazel (*Hamamelis virginiana*) or sweet chestnut (*Castanea sativa*).

Extracts and distillates of witch hazel bark, twigs, and leaves containing **1** are widely used as components of skin care products and in dermatological treatment of sunburn, irritated skin, acne, atopic eczema, and to promote wound healing through anti-inflammatory effects [14,15,16,17,18]. Pure gallotannin **1** inhibits the activity of α-TNF (tumor necrosis factor) [19], autoactivation of plasma hyaluronan-binding protein [20] and exhibits high scavenging and protective activity against cell damage by active oxygen and peroxide [21,22,23]. It also appears to be a promising chemotherapeutic agent, which might be used in the treatment of colon cancer without compromising the viability of healthy colon cells [24]. Very recently, **1** [25,26,27], cyclodextrin–hamamelitannin complexes [28], or different synthetic analogues of hamamelitannin analogues [29,30,31] in combination with antibiotics, were studied as perspective suppressors of staphylococcal infections by inhibiting virulence of bacterial biofilms through quorum sensing mechanisms.

The antiviral efficacy against influenza A virus and human papillomavirus of tannins from *Hamamelis virginiana* bark extract has also been demonstrated [32]. Gallotannin **1** has also become a relatively successful molecule in various in silico screening models aimed at studying the inhibition of proteins important in the process of carcinogenesis, atherosclerosis, or SARS-CoV-2 disease [33,34,35,36].

Despite numerous reports on the medical effects of hamamelitannin, only one total synthesis of **1** has been published so far, as early as 1969 [37]. The authors obtained the target di-*O*-acyl-glycoside in only 22% yield by conventional acylation of the prepared benzyl β-D-hamamelofuranoside with tri-*O*-benzylgalloyl chloride in a pyridine/tetrahydrofuran mixture. Acylation proceeded for 71 h at −40° to rt and afforded three products. The main product was hydrogenated over 10% palladium on charcoal to give compound **1** with a yield of 58%. The starting branched sugar D-hamamelose (2-*C*-hydroxymethyl-D-ribose) was prepared from D-arabinose in several steps via methyl 3,4-*O*-isopropylidene-β-D-*erythro*-pentopyranosidulose [38].

Regioselective acylation of polyhydroxylated molecules like sugars is often a challenge [39]. The solution is to use biocatalysts, especially lipases, and perform enzyme-catalyzed acylation [40,41]. Lipases tolerate a wide range of substrates and are able to work in an aqueous environment as well as in organic solvents [42]. Although they have primarily evolved to hydrolyze triacylglycerols with long fatty acids, some of them also tolerate phenolic substrates, making them similar in reactivity to feruloyl esterases [43]. In aprotic organic solvents, they can catalyze esterifications or transesterifications. The choice of a suitable solvent in reactions is important for their speed as well as selectivity [44,45]. Several reaction steps can be saved by the appropriate selection of the biocatalyst and reaction conditions in the acylation of carbohydrates. They work under mild conditions, so reactions of this type do not consume much energy. Moreover, they are commonly commercially available and can be used multiple times.

In this study, we report two simpler and more efficient syntheses of hamamelitannin **1** from different starting compounds by conventional as well as lipase-promoted galloylation. The regioselectivity of various methods of galloylation of hamamelofuranose, including the enzymatic procedure, was studied.

## 2. Materials and Methods

### 2.1. General

The reactions were performed with commercial reagents purchased from Sigma-Aldrich (Saint-Louis, MI, USA), Acrōs Organics (part of Thermo Fisher Scientific, Waltham, MA, USA), Merck (Darmstadt, Germany), Fluorochem (Hadfield, UK). Molecular sieves with porosity 4Å were microwave-dried before use. Dichloromethane (P_2_O_5_), toluene (Na), acetonitrile (CaH_2_) were dried and distilled before use. D-Hamamelose was a gift from the Production Department of our Institute (Institute of Chemistry SAS, Bratislava, Slovakia). The Lipozyme TL IM, a product of Novozymes (Bagsværd, Denmark), was purchased from Biotech (Trnava, Slovakia). The solvents used in the enzymatic reactions *t*-butanol (*t*-BuOH), acetonitril (CH_3_CN) and *t*-butyl methyl ether (*t*-BuMeO) were of HPLC grade and predried over molecular sieves. Toluene (T), ethyl acetate (EtOAc), dichloromethane (CH_2_Cl_2_), tetrahydrofuran (THF) and methanol (MeOH) were dried (Na, P_2_O_5_, CaH_2_) and distilled before use. All reactions containing sensitive reagents were carried out under the argon atmosphere. TLC was performed on aluminum sheets pre-coated with silica gel 60 F_254_ (Merck, Darmstadt, Germany). The spots were visualized under UV lamp (λ_max_ = 254 nm) and charred with 5% sulfuric acid in ethanol containing 1% orcinol and heating with a heat gun. Column chromatography was performed on Silica gel 60 (0.035–0.070 mm, pore diameter ca. 6 nm, Acrōs Organics). Melting points were recorded with a Kofler hot-block and were uncorrected. Optical rotations were measured on a Jasco P2000 polarimeter at 20 °C. The structures of products were determined by a combination of ^1^H and ^13^C NMR spectroscopy as well as by two-dimensional homonuclear and heteronuclear techniques (COSY, HSQC) recorded on a 400 MHz Bruker AVANCE III HD 400 MHz equipped with a Prodigy CryoProbe. Chemical shifts are reported in ppm (δ) and are referenced to internal CD_3_OD (δ 3.31, for ^1^H and δ 49.00, for ^13^C) or CHCl_3_ (δ 7.26, for ^1^H and δ 77.00, for ^13^C). Scalar couplings are reported in hertz (Hz). High-resolution mass determination was performed on the Orbitrap Velos Pro Thermo Scientific mass analyzer (ion source HESI, capillary temperature 350 °C, source heater temperature 300 °C).

### 2.2. Synthesis of Hamamelitannin from D-Ribose

#### 2.2.1. 2,3-O-Isopropylidene-α,β-D-Hamamelofuranose (**2**)

Potassium carbonate (3.75 g) and an aqueous solution of formaldehyde (37% + 10% MeOH) (50 mL) were dissolved in methanol (75 mL), and 2,3-*O*-isopropylidene-α,β-D-ribofuranose [46] (9.51 g, 50 mmol) was added to the reaction solution. The reaction mixture was stirred at 80 °C under argon for 40 h, after which time it was neutralized with 1M H_2_SO_4_. Evaporation to dryness gave a residue that was extracted with hot ethyl acetate (3 × 100 mL), dried (Na_2_SO_4_), and the extracts were concentrated to a syrup. Purification of the crude product by column chromatography on silica gel (toluene:ethyl acetate, 2:1 → 0:1) gave **3** as a homogeneous syrup (8.88 g, 81%, α:β = 1:0.6); [α]_D_^20^ = +9.6° (c = 1.0, CH_3_OH), (lit. [47] [α]_D_^23^ = +9.3° (c = 3.0, H_2_O)). ^1^H NMR (400 MHz, CDCl_3_) δ: 5.42 (d, H-1α, transformed into a singlet on addition of D_2_O), 5.29 (d, H-1β, transformed into a singlet on addition of D_2_O), 4.58 (s, H-3α), 4.59 (d, *J* = 3.8 Hz, H-3β), 4.32 (dt, *J* = 4.3, 3.3, 1.1 Hz, H-4α), 4.24 (bdd, *J* = 3.3, 1.1 Hz, H-4β), 3.91 (d, *J* = 12.0 Hz, H-2′aα), 3.83 (d, *J* = 12.0 Hz, H-2′bα), 3.82 (s, H-2′aβ, H-2′bβ), 3.79–3.67 (m, H-5aβ, H-5bβ, H-5aα, H-5bα), 1.59 (s, CH_3_β), 1.50 (s, CH_3_α), 1.46 (s, CH_3_β),1.44 (s, CH_3_α). ^13^C NMR (101 MHz, CDCl_3_) δ: 114.9 (CMe_2_β), 113.5 (CMe_2_α), 103.6 (C-1α), 98.1 (C-1β), 94.5 (C-2α), 91.1 (C-2β), 87.5 (C-4α), 84.0 (C-3α), 83.2 (C-3β), 82.5 (C-4β), 63.3 (C-5β, C-5α), 62.8 (C-2′β), 62.7 (C-2′α), 28.1 (CH_3_α), 27.6 (CH_3_α), 27.1 (CH_3_β), 27.0 (CH_3_β). HRMS (ESI): *m/z* calcd for C_9_H_16_O_6_Na ([M + Na]^+^) 243.08446; found, 243.08389.

#### 2.2.2. 3,4,5-Tri-O-Acetylgalloyl Chloride (**3**)

To vigorously stirred gallic acid (8.51 g, 50 mmol) in Ac_2_O (20 mL), 3 drops of concentrated H_2_SO_4_ were added at 5 °C. The mixture was then stirred for 60 min at room temperature and poured into an ice/water mixture (200 mL). After 2 h at room temperature, the precipitated white solid was washed with water until the filtrate was neutral, then filtered, dried, and crystallized (ethanol) to afford 12.56 g (85%) of 3,4,5-triacetoxybenzoic acid; Mp: 154–157 °C. To the acetylated gallic acid (5.92 g, 20 mmol) in toluene (80 mL), thionyl chloride (7.4 mL, 100 mmol) was added. The mixture was stirred for 3 h at 70 °C. After the evaporation of the liquids to one-third of the original volume, the mixture was cooled and the precipitated solid was washed with cyclohexane and dried. White solid (5.91 g, 94%) was obtained; Mp 104–105 °C.

#### 2.2.3. Acylation Methods for Galloylation of 2 (Table 1)

Method A: Hamamelofuranose **2** (0.22 g, 1 mmol) and 3, 4,5-tri-*O*-acetylgalloyl chloride **3** (1.04 g, 3.3 mmol) were dissolved in dry CH_2_Cl_2_ (10 mL). Triethylamine (0.139 mL, 1 mmol) and 4-dimethylaminopyridine (0.031 g, 0.25 mmol) were added at 0 °C. The reaction mixture was then stirred for 3 h at laboratory temperature. The mixture was then diluted with CH_2_Cl_2_ (40 mL), washed with 1% HCl (10 mL), water (2 × 20 mL), dried over Na_2_SO_4_, and concentrated under reduced pressure. Products were isolated by column chromatography of the residue on silica gel (toluene/EtOAc, 2:1 → 1:2).

Method B: Hamamelofuranose **2** (0.22 g, 1 mmol) and dibutyltin oxide (0.548 g, 2.2 mmol) were dissolved in dry methanol (10 mL) and refluxed for 2 h. After evaporation of the solvent, the residue was dried under vacuum, dissolved in dichloromethane (10 mL), and cooled to 0 °C. The acylation reagent **2** (0.693 g, 2.2 mmol) in dichloromethane (6 mL) was added dropwise and then allowed to react at room temperature for 2 h. The resulting mixture was concentrated and directly purified by chromatography (toluene/EtOAc, 2:1 → 1:2).

Method C: Hamamelofuranose **2** (0.22 g, 1 mmol) and triacetylgalloyl chloride **3** (1.04 g, 3.3 mmol) were dissolved in dry CH_3_CN (5 mL). Zinc oxide (0.9 g, 11 mmol) was added in one portion. The heterogeneous mixture was stirred for 24 h at 40 °C, then diluted with ethyl acetate (10 mL) and filtered through Celite 545. The residue after concentration of the filtrate under reduced pressure was purified by column chromatography on silica gel (toluene/EtOAc, 2:1 → 1:2).

Method D: Isopropylidenated hamamelose **2** (0.220 g, 1 mmol) and vinyl 3,4,5-tri-*O*-acetylgallate (**4a**) [48] (0.588 g, 3.0 mmol, 3.0 equiv.) were dissolved in dry *t*-BuMeO (20 mL) at room temperature. Activated molecular sieves 4Å (0.5 g) and Lipozyme TL IM (0.4 g) were added and the reaction mixture was shaken at 450 rpm and 37 °C for 19 h. The reaction was stopped by filtration; the filter cake was washed with ethyl acetate, and combined organic phases were concentrated under reduced pressure. The residue was purified by chromatography on a silica gel column eluted with toluene/EtOAc (2:1 → 1:2) to afford an unexpected product—2,3-*O*-isopropylidene-2′-*O*-acetyl-5-*O*-(3,4,5-tri-*O*-acetylgalloyl)-α,β-D-hamamelofuranose (**5a**-2′-Ac) (34%), 2,5-diacyl **6a** (6%), and 5-monoacyl **5a** (9%).

Method E: Isopropylidenated hamamelose **2** (0.220 g, 1 mmol) and vinyl gallate (**4b**) [48] (0.588 g, 3.0 mmol, 3.0 equiv.) were dissolved in dry *t*-BuOH (10 mL) or *t*-BuMeO (20 mL) at room temperature. Activated molecular sieves 4Å (0.5 g) and Lipozyme TL IM (0.8 g) were added and the reaction mixture was shaken at 450 rpm and 37 °C for 242 h (in *t*-BuOH) or 61 h (in *t*-BuMeO) and then finished by filtration. The filter cake was washed several times with ethyl acetate and combined organic phases were concentrated under reduced pressure. The residue was purified by chromatography on the column of silica-gel eluted with toluene/EtOAc (1:2 → 1:5) to afford 5-*O*-gallate **5b** (0.245 g, 66% in *t*-BuOH or 0.304 g, 82% in *t*-BuMeO) as an amorphous white solid.
biomolecules-13-00519-t001_Table 1Table 1Acylation of **2** by acyl donors under various conditions.EntryMethod ^1^Acyl Donor/Equiv.Catalyst(Equiv.)SolventTemp.(°C)Time(h)5a ^2^(%)6a(%)7(%)1A**3**/2.22.0/0.5CH_2_Cl_2_0-rt32422482A**3**/3.33.0/0.75CH_2_Cl_2_0-rt349813B**3**/2.21.0CH_2_Cl_2_0-rt2.5194484B**3**/2.22.2CH_3_CN5063522n.d. ^3^5B**3**/2.22.2CH_2_Cl_2_0-rt258446C**3**/2.24.9CH_2_Cl_2_rt124921
7C**3**/2.24.9CH_3_CN404712218C**3**/3.311CH_3_CN4024956279D**4a**/3.0-*t*-BuMeO37199 + 34 ^4^6n.d.^1^ Method A—catalyst: Et_3_N/DMAP; method B—catalyst: Bu_2_SnO; method C—catalyst: ZnO; method D—biocatalyst: Lipozyme TL IM. ^2^ The yield of isolated monoacylated products mixture. ^3^ Not detected. ^4^ Position 2′-OH was acetylated.


#### 2.2.4. Characterization Data of Acylated Products

2,3-*O*-Isopropylidene-5-*O*-(3,4,5-tri-*O*-acetylgalloyl)-α,β-D-hamamelofuranose (**5a**) from the enzymatic reaction (Method D). Colourless foam; α:β = 1:0.8; [α]_D_^20^ = +5.0 (c = 1.0, CH_3_OH). ^1^H NMR (400 MHz, CD_3_OD) δ: 7.85 (2xs, 4H, H-Arα, H-Arβ), 5.36 (s, 1H, H-1α), 5.19 (s, 1H, H-1β), 4.65 (bs, 2H, H-3α, H-3β), 4.51–4.33 (m, 6H, H-4α, H-4β, H-5aβ, H-5bβ, H-5aα, H-5bα), 3.84 (bd, *J* = 1.3 Hz, 2H, H-2′aα, H-2′bα), 3.73 (d, 1H, *J* = 11.8 Hz, H-2′aβ), 3.66 (d, 1H, *J* = 11.9 Hz, H-2′bβ), 2 × 2.32 (s), 4 × 2.32 (s) (6 × CH_3_CO), 1.58 (s, CH_3_β), 1.50 (s, CH_3_α), 1.46 (2 × s, CH_3_αβ). ^13^C NMR (101 MHz, CD_3_OD) δ: 169.4 (4 × COCH_3_), 168.2 (COOβ, COOα), 165.6 (2 × COCH_3_), 145.0 (2 × C-Ar), 140.4 (C-Ar), 129.2 (C-Arα), 129.1 (C-Arβ), 123.3 (2 × CH-Arαβ), 116.2 (CMe_2_β), 114.8 (CMe_2_α), 105.5 (C-1α), 99.1 (C-1β), 95.8 (C-2α), 92.6 (C-2β), 85.7 (C-3α), 84.7 (C-4α), 83.9 (C-3β), 80.8 (C-4β), 67.3 (C-5α), 65.7 (C-5β), 65.4 (C-2′β), 63.0 (C-2′α), 28.3 (2 × CH_3_α), 27.8 (CH_3_β), 27.4 (CH_3_β), 20.4 (4 × COCH_3_), 20.0 (2 × COCH_3_). HRMS (ESI): *m/z* calcd for C_22_H_26_O_13_Na ([M + Na]^+^) 521.12656; found 521.12687.

2,3-*O*-Isopropylidene-2′-*O*-acetyl-5-*O*-(3,4,5-tri-*O*-acetylgalloyl)-α,β-D-hamamelofuranose (**5a**-2′-Ac). Colorless foam; α:β = 1:0.5; [α]_D_^20^ = +7.0 (c = 1.0, CH_3_OH). ^1^H NMR (400 MHz, CD_3_OD) δ: 7.83 (s, 2H, H-Arα), 7.79 (s, 2H, H-Arβ), 5.34 (s, 1H, H-1α), 5.15 (s, 1H, H-1β), 4.69 (d, *J* = 1.2 Hz, 1H, H-3α), 4.58 (d, 1H, *J* = 2.1 Hz, H-3β), 4.52–4.38 (m, 8H, H-4α, H-4β, H-5aβ, H-5bβ, H-5aα, H-5bα, H-2′aα, H-2′bα), 4.37 (d, 1H, *J* = 11.9 Hz, H-2′aβ), 4.19 (d, 1H, *J* = 11.9 Hz, H-2′bβ), 2 × 2.30 (s), 4 × 2.29 (s) (6 × CH_3_CO), 2.07 (s, 3H, COCH_3_α), 2.06 (s, 3H, COCH_3_β), 1.57 (s, CH_3_β), 1.48 (s, CH_3_α), 1.43 (s, CH_3_β),1.42 (s, CH_3_α). ^13^C NMR (101 MHz, CD_3_OD) δ: 172.5 (COCH_3_), 172.1 (COCH_3_), 169.4 (4 × COCH_3_), 168.2 (ArCOOβ, ArCOOα), 165.6 (2 × COCH_3_), 145.1 (2 × C-Arβ), 145.0 (2 × C-Arα), 140.4 (C-Arβ), 140.4 (C-Arα), 129.2 (C-Arα), 128.9 (C-Arβ), 123.3 (CH-Arα), 123.2 (CH-Arβ), 117.0 (CMe_2_β), 115.2 (CMe_2_α), 104.9 (C-1α), 99.0 (C-1β), 94.1 (C-2α), 91.2 (C-2β), 86.0 (C-3α), 84.7 (C-4α), 83.7 (C-3β), 80.9 (C-4β), 67.3 (C-5α), 65.8 (C-5β), 65.3 (C-2′β), 64.9 (C-2′α), 28.2 (2 × CH_3_α), 27.6 (CH_3_β), 27.2 (CH_3_β), 20.8 (COCH_3_), 20.7 (COCH_3_), 20.5 (COCH_3_), 20.4 (3 × COCH_3_), 20.0 (2 × COCH_3_). HRMS (ESI): *m/z* calcd for C_24_H_28_O_14_Na ([M + Na]^+^) 563.13768; found 563.13758.

2,3-*O*-Isopropylidene-5-*O*-galloyl-α,β-D-hamamelofuranose (**5b**) from enzyme reaction (Method E). Colorless foam; α:β = 1:0.7; [α]_D_^20^ = +8.7° (c = 1.0, CH_3_OH). ^1^H NMR (400 MHz, CD_3_OD) δ: 7.08 (s, 2H, H-Arα), 7.07 (s, 2H, H-Arβ), 5.34 (s, 1H, H-1α), 5.19 (s, 1H, H-1β), 4.62 (s, 1H, H-3α), 4.58 (d, 1H, *J* = 1.8 Hz, H-3β), 4.42–4.25 (m, 6H, H-4α, H-4β, H-5aβ, H-5bβ, H-5aα, H-5bα), 3.85 (d, 1H, *J* = 12.3 Hz, H-2′aα), 3.81 (d, 1H, *J* = 12.3 Hz, H-2′bα), 3.73 (d, 1H, *J* = 11.8 Hz, H-2′aβ), 3.68 (d, 1H, *J* = 11.8 Hz, H-2′bβ), 1.56 (s, CH_3_β), 1.48 (s, CH_3_α), 1.44 (s, CH_3_β),1.43 (s, CH_3_α). ^13^C NMR (101 MHz, CD_3_OD) δ: 168.0 (COOβ), 168.0 (COOα), 146.5 (2 × C-Arβ), 146.5 (2 × C-Arα), 140.0 (C-Arβ), 140.0 (C-Arα), 121.2 (C-Arα), 121.1 (C-Arβ), 116.2 (CMe_2_β), 114.8 (CMe_2_α), 110.2 (2 × CH-Ar), 105.4 (C-1α), 99.2 (C-1β), 95.8 (C-2α), 92.6 (C-2β), 86.0 (C-3α), 84.9 (C-4α), 84.4 (C-3β), 81.0 (C-4β), 66.3 (C-5α), 64.8 (C-5β), 63.6 (C-2′β), 63.2 (C-2′α), 28.3 (2 × CH_3_α), 27.7 (CH_3_β), 27.4 (CH_3_β). HRMS (ESI): *m/z* calcd for C_16_H_20_O_10_Na ([M + Na]^+^) 395.09487; found 395.09497.

2,3-*O*-Isopropylidene-2′,5-di-*O*-(3,4,5-tri-*O*-acetylgalloyl)-α,β-D-hamamelofuranose (**6a**). Colorless foam; α:β = 1:0.9, [α]_D_^20^ = −7.9° (c = 1.0, CHCl_3_). ^1^H NMR (400 MHz, CDCl_3_) δ: 7.83 (3 × s, H-Ar), 7.81 (s, H-Ar), 5.51 (d, J_1,OH_ = 2.2 Hz, H-1α), 5.33 (dd, J_1,OH_ = 9.3 Hz, H-1β), 4.71–4.37 (m, H-3α, H-3β, H-4α, H-4β, H-5aβ, H-5bβ, H-5aα, H-5bα, H-2′aα, H-2′bα, H-2′aβ, H-2′bβ), 3.82 (d, OHβ), 3.55 (d, OHα), 2 × 2.30, 3 × 2.29, 2.28, (6s, 6 × CH_3_CO), 1.61 (s, CH_3_β), 1.50 (s, CH_3_α), 1.47 (s, CH_3_β),1.42 (s, CH_3_α). ^13^C NMR (101 MHz, CDCl_3_) δ: 2 × 167.5, 166.3, 164.3, 164.1, 163.9, (COCH_3_), 143.5 (2 × C-Ar), 143.4 (2 × C-Ar), 138.9 (C-Ar), 128.0 (C-Ar), 127.7 (C-Ar), 127.5 (C-Ar), 127.3 (C-Ar), 122.4 (3 × CH-Ar), 122.2 (CH-Ar), 116.1 (CMe_2_), 114.6 (CMe_2_), 103.5 (C-1α), 98.1 (C-1β), 93.1, 89.3 (C-2α, C-2β), 84.6, 84.5, 83.5, 79.5, (C-3α, C-3β, C-4α, C-4β), 66.2, 65.1, 65.0, 64.4 (C-5α, C-5β, C-2′α, C-2′β), 28.2 (CH_3_), 27.7 (CH_3_), 27.3 (2 × CH_3_), 20.5 (COCH_3_), 20.1 (COCH_3_). HRMS (ESI): *m/z* calcd for C_35_H_36_O_20_Na ([M + Na]^+^) 799.16976; found 799.17012.

From enzyme reaction (Method D) **6a**, white solid, α:β = 1:0.5. ^1^H NMR (400 MHz, CD_3_OD, 40 °C) δ: 7.84 (s, H-Arβ), 7.83 (s, H-Arα), 7.82 (s, H-Arα), 7.80 (s, H-Arβ), 5.42 (s, H-1α), 5.26 (s, H-1β), 4.85 (d, *J* = 1.2 Hz, H-3α), 4.76 (d, *J* = 1.8 Hz, H-3β), 4.71 (d, *J* = 12.1 Hz, H-2′aα), 4.61 (d, *J* = 12.1 Hz, H-2′bα), 4.57–4.36 (m, H-4α, H-4β, H-5aβ, H-5bβ, H-5aα, H-5bα, H-2′aβ, H-2′bβ), 2 × 2.29 (s), 2 × 2.29 (s) 2.28 (s), 2 × 2.27 (s), 2.26 (s), 4 × 2.26 (s) (12 × CH_3_CO), 1.58 (s, CH_3_β), 1.49 (s, CH_3_α), 1.40 (s, CH_3_β),1.47 (s, CH_3_α). ^13^C NMR (101 MHz, CD_3_OD) δ: 8 × 169.4, 4 × 168.2, 3 × 165.7, 165.4 (12 × COCH_3_ and 4 × ArCOO), 4 × 145.1, 140.5 (C-Ar), 140.4 (3 × C-Ar), 129.3 (C-Arα), 129.2 (C-Arα), 129.0 (C-Arβ), 129.9 (C-Arβ), 123.3 (CH-Arβ), 123.3 (2 × CH-Arα), 123.2 (CH-Arβ), 117.1 (CMe_2_β), 115.4 (CMe_2_α), 105.0 (C-1α), 99.3 (C-1β), 94.2, 91.3 (C-2α, C-2β), 84.9 (C-3α), 84.5 (C-5α), 83.5 (C-3β), 79.5 (C-5β), 67.3 (C-5α), 66.0 (C-2′α,), 66.4, 65.9 (C-5β, C-2′β), 28.4 (CH_3_α), 28.1 (CH_3_α), 27.8 (CH_3_β), 27.2 (CH_3_β), 20.4 (COCH_3_), 20.0 (COCH_3_). HRMS (ESI): *m/z* calcd for C_35_H_36_O_20_Na ([M + Na]^+^) 799.16976; found 799.16946.

2,3-*O*-Isopropylidene-1,2′,5-tri-*O*-(3,4,5-tri-*O*-acetylgalloyl)-α-D-hamamelofuranose (**7**). White solid, mp 129–130 °C (EtOH); [α]_D_^20^ = −29.1° (c = 1.0, CHCl_3_). ^1^H NMR (400 MHz, CDCl_3_) δ: 7.80 (s, 2H, H-Ar), 7.79 (s, 2H, H-Ar), 7.73 (s, 2H, H-Ar), 6.65 (s, 1H, H-1), 4.78 (bs, 1H, H-3), 4.77 (d, *J* = 12.2 Hz, 1H, H-2′a), 4.68–4.60 (m, 2H, H-4, H-2′b), 4.48 (bd, *J* = 7.4 Hz, 2H, H-5a, H-5b), 2.29, 2.28, 2.28, 2.26 (4s, 27H, 9 × CH_3_CO), 1.55 (s, 3H, CH_3_), 1.44 (s, 3H, CH_3_). ^13^C NMR (101 MHz, CDCl_3_) δ: 167.5, 167.4, 166.2, 166.1, 163.8, 162.6 (9 × COCH_3_), 143.6 (C-Ar), 143.5 (2 × C-Ar), 139.3 (C-Ar), 139.0 (2 × C-Ar), 127.4 (2 × C-Ar), 127.0 (C-Ar), 122.4 (2 × CH-Ar), 122.2 (CH-Ar), 115.2 (CMe_2_), 102.9 (C-1), 92.9 (C-2), 85.3 (C-4), 84.2 (C-3), 64.7 (C-5), 64.4 (C-2′), 27.8 (CH_3_), 27.6 (CH_3_), 20.5 (COCH_3_), 20.1 (COCH_3_). HRMS (ESI): *m/z* calcd for C_48_H_46_O_27_Na ([M + Na]^+^) 1077.21242; found 1077.21210.

### 2.3. Enzymatic Acylation of D-Hamamelose

#### 2.3.1. Preparation of New Derivatives of Vinyl Gallate

Vinyl 3,4,5-tri-*O*-(*t*-butyldimethylsilyl)gallate (**4f**). Vinyl gallate (0.690 g, 3.5 mmol) and *t*-butyldimethylsilyl chloride (1.662 g, 11.03 mmol) were dissolved in dry THF (5 mL), then Et_3_N (1.61 mL, 11.6 mmol) and DMAP (0.32 g, 2.63 mmol) were added at 0 °C. The reaction mixture was stirred for 1 h at rt., then toluene was added and the precipitated salts were filtered off and washed with toluene. After concentration of the filtrate, the product was purified by flash chromatography (toluene). Ester **4f** (1.53 g, 81%) was obtained as a colorless oil. ^1^H NMR (400 MHz, CDCl_3_,) δ 7.46 (dd, *J* = 14.0, 6.3 Hz, 1H, CH=), 7.27 (s, 2H, CH-gal), 4.99 (dd, *J* = 14.0, 1.6 Hz, 1H, =CH_2_a), 4.65 (dd, *J* = 6.3, 1.6 Hz, 1H, =CH_2_b), 0.99 (s, 9H, H-*t*-Bu), 0.96 (s, 18H, 2 × H-*t*-Bu), 0.25 (s, 12H, 4 × CH_3_), 0.15 (s, 6H, 2 × CH_3_). ^13^C NMR (101 MHz, CDCl_3_) δ 163.3 (COO), 148.6 (2 × C-Ar), 144.0 (C-Ar), 141.5 (CH=), 120.6 (C-Ar), 115.9 (CH-gal), 97.5 (=CH_2_), 26.2 (2 × C(CH_3_)_3_), 26.1 (C(CH_3_)_3_), 18.8 (2 × C(CH_3_)_3_), 18.5 (C(CH_3_)_3_), -3.7 (4 × CH_3_), -3.9 (2 × CH_3_). HRMS (ESI): *m/z* calcd for C_27_H_50_O_5_ Si_3_+H ([M + H]^+^) 539.30388; found 539.30400.

Vinyl 3,4,5-tri-*O*-benzylgallate (**4g**). 3,4,5-Tri-*O*-benzylgallic acid [49] (13.22 g, 30 mmol) was dissolved in dry THF (30 mL) and vinyl acetate (45 mL). Mercury (II) acetate (0.30 g) and BF_3_.OEt_2_ (10 drops) were added to the mixture. The reaction mixture was stirred for 4 h at 37 °C, then neutralized with anhydrous CH_3_COONa. After filtration, the mixture was concentrated and purified by chromatography (CHCl_3_). A white solid 9.38 g (67%) was obtained, mp 108 °C. ^1^H NMR (400 MHz, CDCl_3_,) δ 7.46 (dd, *J* = 14.1, 6.5 Hz, 1H, CH=), 7.45–7.42 (m, 4H, H-Ph), 7.43 (s, 2H, CH-gal), 7.40–7.31 (m, 8H, H-Ph), 7.28–7.22 (m, 3H, H-Ph), 5.14 (bs, 4H, 2 × CH_2_), 5.13 (s, 2H, CH_2_), 5.05 (dd, *J* = 14.0, 1.7 Hz, 1H, =CH_2_a), 4.69 (dd, *J* = 6.2, 1.7 Hz, 1H, =CH_2_b). ^13^C NMR (101 MHz, CDCl_3_) δ 163.2 (COO), 152.6 (2 × C-Ar), 143.1 (C-Ar), 141.5 (CH=), 137.3 (C-Ar), 136.5 (2 × C-Ar), 128.5 (4 × CH-Ph), 128.5 (2 × CH-Ph), 128.2 (2 × CH-Ph), 128.1 (2 × CH-Ph), 128.0 (CH-Ph), 127.6 (4 × CH-Ph), 123.8 (C-Ar), 109.5 (CH-gal), 98.2 (=CH_2_), 75.1 (2 × CH_2_), 71.3 (CH_2_). HRMS (ESI): *m/z* calcd for C_30_H_26_O_5_Na ([M + Na]^+^) 489.16725; found 489.16728.

#### 2.3.2. Enzymatic Benzoylation of **8**

D-Hamamelose (0.36 g, 2 mmol) was suspended in *t*-BuMeO (40 mL). Molecular sieves 4Å (2g), vinyl benzoate (1.11 mL, 4 equiv.), and Lipozyme TL IM (0.4 g) were added. The tightly closed reaction mixture was shaken on a vibrating shaker at 450 rpm in an incubator at 37 °C. After 50 h, the reaction was filtered through Celite 545, the filter cake was washed several times with EtOAc, and the filtrate was concentrated. The reaction mixture was purified on a silica gel column eluted with toluene/EtOAc (3:1→1:2). Several products were obtained during the elution in the order: **13** (3%), **11c** (35%), **12c** (17%), **9c** (37%), and **10c** (1%).

2′,5-Di-*O*-benzoyl-α,β-D-hamamelofuranose (**9c**). White solid, mp 147–149 °C; α:β = 0.75:1; [α]_D_^20^ = +37.7° (c = 1.0, CH_3_OH). ^1^H NMR (400 MHz, CD_3_OD) δ: 8.08 (dt, *J* = 8.4, 1.2 Hz, H-Ph), 8.01 (td, *J* = 8.2, 1.4 Hz, H-Ph), 7.63–7.54 (m, H-Ph), 7.49–7.36 (m, H-Ph), 5.35 (s, H-1α), 5.26 (s, H-1β), 4.64 (dd, *J* = 12.2, 2.8 Hz, 5aα), 4.61 (dd, *J* = 11.9, 2.7 Hz, H-5aβ), 4.56 (d, *J* = 11.5 Hz, 2′aβ), 4.50 (d, *J* = 11.5 Hz, 2′bβ), 4.47–4.41 (m, H-5bα, H-5bβ), 4.39 (d, *J* = 11.5 Hz, 2′aα), 4.34 (d, *J* = 11.5 Hz, 2′bβ), 4.29 (ddd, *J* = 7.7, 4.8, 2.7 Hz, H-4α), 4.27–4.20 (m, H-3β, H-4β), 4.08 (d, *J* = 8.1 Hz, H-3α). ^13^C NMR (101 MHz, CD_3_OD) δ 168.3 (COOβ), 168.1 (COOβ), 167.9 (COOα), 167.7 (COOα), 134.4 (CH-Phα), 134.3 (CH-Phα), 134.2 (CH-Phβ), 134.2 (CH-Phβ), 131.5 (C-Phβ), 131.4 (C-Phβ), 131.2 (C-Phα), 131.1 (C-Phα), 130.7 (2 × CH-Phβ), 130.7 (2 × CH-Phβ), 130.64 (2 × CH-Phα), 130.5 (2 × CH-Phα), 129.6 (2 × CH-Phα), 129.6 (2 × CH-Phα), 129.5 (2 × CH-Phβ), 129.5 (2 × CH-Phβ), 103.0 (C-1β), 98.9 (C-1α), 81.4 (C-4β), 80.7 (C-2β), 80.4 (C-4α), 77.7 (C-2α), 74.1 (C-3β), 72.5 (C-3α), 67.7 (C-2′β), 66.9 (C-5β), 66.4 (C-2′α), 65.1(C-5α). HRMS (ESI): *m/z* calcd for C_20_H_20_O_8_Na ([M + Na]^+^) 411.10504; found 105.10543.

2′,3,5-Tri-*O*-benzoyl-α,β-D-hamamelofuranose (**11c**) and 1,2′,5-tri-*O*-benzoyl-α-D- hamamelofuranose (**12c**α). White amorphous solid; **12**α:**11**α:**11**β = 0.4:0.6:1; [α]_D_^20^ = +25.8° (c = 1.0, CH_3_OH). ^1^H NMR (400 MHz, CD_3_OD) δ: 8.17–7.88 (m, H-Ph), 7.68–7.18 (m, H-Ph), 6.54 (s, H-1α-**12**), 5.82 (d, *J* = 7.1 Hz, H-3β-**11**), 5.52 (d, *J* = 6.8 Hz, H-3α-**11**), 5.40 (s, H-1α-**11**), 5.32 (s, H-1β-**11**), 4.71–4.45 (m, H-4α-**11**, H-4β-**11**, H-5aα-**11**, H-5aβ-**11**, H-5aα-**12**, H-5bα-**11**, H-5bβ-**11**, H-5bα-**12**, H-2′aβ-**11**, H-2′aα-**11**, H-2′aα-**12**, H-2′bβ-**11**, H-4α-**12**, H-2′bα-**11**), 4.42 (d, *J* = 11.6 Hz, H-2′bα-**12**), 4.18 (d, *J* = 6.6 Hz, H-3α-**12**). ^13^C NMR (101 MHz, CD_3_OD) δ 167.9 (COO), 167.7 (COO), 167.7 (2 × COO), 167.6 (COO), 167.6 (COO), 167.3 (COO), 167.3 (COO), 166.8 (COO), 134.6, 134.6, 134.5, 134.4, 134.4, 134.3, 134.3, 134.2, 134.1, 134.0 (CH-Ph), 3 × 131.0, 131.0, 131.0, 130.9, 130.9, 130.9, 130.8, 130.8, 130.7, 130.7, 3 × 130.7, 2 × 130.6, 130.6, 130.5 (CH-Ph and C-Ph), 129.6, 129.6, 2 × 129.6, 3 × 129.5, 129.4, 129.4, 129.3 (CH-Ph), 103.7 (C-1β-**11**), 99.4 (C-1α-**12**), 98.7 (C-1α-**11**), 83.9 (C-4α-**12**), 81.2 (C-2β-**11**), 79.1 (C-4β-**11**), 3 × 78.9 (C-4α-**11**, C-2α-**11**, C-2α-**12**), 77.1 (C-3β-**11**), 74.1 (C-3α-**11**), 71.5 (C-3α-**12**), 68.2 (C-2′β-**11**), 67.4 (C-2′α-**11**), 67.3 (C-2′α-**12**), 66.8 (C-5β-11), 65.1 (C-5α-**11**), 64.7 (C-5α-**12**). HRMS (ESI): *m/z* calcd for C_27_H_24_O_9_Na ([M + Na]^+^) 515.13125; found 515.13167.

1,2′,5-Tri-*O*-benzoyl-β-D-hamamelofuranose (**12c**β). White solid, mp 126–128 °C; [α]_D_^20^ = −17.9° (c = 1.0, CH_3_OH). ^1^H NMR (400 MHz, CD_3_OD) δ 7.94–7.79 (m, 6H, H-Ph), 7.57–7.17 (m, 9H, H-Ph), 6.38 (s, 1H, H-1), 4.73 (dd, *J* = 12.4, 2.7 Hz, 1H, H-5a), 4.69 (d, *J* = 11.8 Hz, 1H, H-2′a), 4.56 (d, *J* = 8.4 Hz, 1H, H-3), 4.55 (d, *J* = 11.8 Hz, 1H, H-2′b), 4.47 (dd, *J* = 12.4, 3.7 Hz, 1H, H-5b), 4.41 (ddd, *J* = 8.4, 3.7, 2.7 Hz, 1H, H-4). ^13^C NMR (101 MHz, CD_3_OD) δ 167.7 (COO), 167.6 (COO), 166.3 (COO), 134.5 (CH-Ph), 134.3 (CH-Ph), 134.2 (CH-Ph), 130.9 (C-Ph), 130.9 (C-Ph), 130.6 (4 × CH-Ph), 130.6 (C-Ph), 130.5 (2 × CH-Ph), 129.6 (2 × CH-Ph), 129.5 (4 × CH-Ph), 101.6 (C-1), 82.8 (C-4), 80.7 (C-2), 72.3 (C-3), 66.9 (C-2′), 64.1 (C-5). HRMS (ESI): *m/z* calcd for C_27_H_24_O_9_Na ([M + Na]^+^) 515.13125; found 515.13155.

1,2′,4-Tri-*O*-benzoyl-D-hamamelopyranose (**13**). Amorphous white solid. ^1^H NMR (400 MHz, CD_3_OD) δ 8.18–7.95 (m, 6H, H-Ph), 7.64–7.39 (m, 9H, H-Ph), 6.50 (s, 1H, H-1), 5.50–5.43 (m, 1H, H-4), 4.81 (d, *J* = 11.8 Hz, 1H, H-2′a), 4.48 (d, *J* = 4.0 Hz, 2H, H-3), 4.47 (d, *J* = 11.8 Hz, 1H, H-2′b), 4.25 (dd, *J* = 13.1, 2.3 Hz, 1H, H-5a), 4.07 (dd, *J* = 13.0, 3.2 Hz, 1H, H-5b). ^13^C NMR (101 MHz, CD_3_OD) δ 167.8 (COO), 167.6 (COO), 166.9 (COO), 134.9 (CH-Ph), 134.4 (CH-Ph), 134.3 (CH-Ph), 131.3 (C-Ph), 131.0 (2 × CH-Ph), 131.0 (C-Ph), 130.8 (2 × CH-Ph), 130.7 (2 × CH-Ph), 130.4 (C-Ph), 129.7 (2 × CH-Ph), 129.5 (2 × CH-Ph), 129.4 (2 × CH-Ph), 95.5 (C-1), 73.9 (C-2), 71.8 (C-4), 67.4, 67.4 (C-3, C-2′), 63.9 (C-5).

#### 2.3.3. Enzymatic Acylation of **8** by Vinyl Gallates **4d–g**

D-Hamamelose **8** (0.18 g, 1 mmol) was suspended in *t*-BuMeO (20 mL) or in *t*-BuOH (20 mL). Molecular sieves 4Å (1g), derivatized vinyl gallate (3 equiv.) and Lipozyme TL IM (0.2 g) were added. The reaction mixture was shaken on a vibrating shaker at 450 rpm in an incubator at 37 °C. After the time indicated in Table 2, the reaction was filtered through Celite 545, the filter cake was washed several times with EtOAc, and the filtrate was concentrated. The reaction mixture was purified on a silica gel column eluted with toluene/EtOAc (3:1→1:2). The data of the products are presented in Table 2.

#### 2.3.4. Characterization Data of Acylated Hamameloses

2′,5-Di-*O*-syringoyl-α,β-D-hamamelofuranose (**9d**). White foam; α:β = 0.4:1; [α]_D_^20^ = +27.3° (c = 1.0, CH_3_OH). ^1^H NMR (400 MHz, CD_3_OD) δ: 7.43 (s, H-Arβ), 7.39 (s, H-Arβ), 7.26 (s, H-Arα), 7.24 (s, H-Arα), 5.31 (s, H-1α), 5.27 (s, H-1β), 4.56 (dd, *J* = 12.1, 2.9 Hz, 5aα), 4.53–4.46 (m, H-5aβ, H-2′aβ, H-2′bβ, H-5bα), 4.42 (dd, *J* = 11.9, 4.6 Hz, H-5bβ), 4.37 (d, *J* = 11.4 Hz, H-2′aα), 4.33 (d, *J* = 7.8 Hz, H-3β), 4.31–4.27 (m, H-4α), 4.28 (d, *J* = 11.4 Hz, 2′bα), 4.23 (ddd, *J* = 7.6, 4.5, 2.8 Hz, H-4β), 4.04 (d, *J* = 7.6 Hz, H-3α), 3.88 (s, OCH_3_β), 3.87 (s, OCH_3_β), 3.84 (s, OCH_3_α), 3.81 (s, OCH_3_α). ^13^C NMR (101 MHz, CD_3_OD) δ 168.2 (COOβ), 168.1 (COOβ), 167.8 (COOα), 167.6 (COOα), 148.9 (2 × C-OCH_3_β), 148.8 (2 × C-OCH_3_α), 142.0 (C-OHβ), 141.9 (C-OHα), 121.4 (C-Arβ), 121.3 (C-Arβ), 120.9 (C-Arα), 120.9 (C-Arα), 108.4 (CH-Arβ), 108.3 (CH-Arβ), 108.2 (CH-Arα), 108.0 (CH-Arα), 102.9 (C-1β), 99.2 (C-1α), 81.6 (C-4β), 81.1 (C-4α), 80.8 (C-2β), 77.9 (C-2α), 73.6 (C-3β), 72.5 (C-3α), 67.7 (C-2′β), 66.9 (C-2′α), 65.8 (C-5β), 64.9 (C-5α), 56.8 (2 × O*C*H_3_β), 56.7 (O*C*H_3_α), 56.6 (O*C*H_3_α). HRMS (ESI): *m/z* calcd for C_24_H_28_O_14_Na ([M + Na]^+^) 563.13713; found 563.13708.

Mono-*O*-syringoyl-α,β-D-hamamelofuranoses (**10d**). Colourless waxy solid; [α]_D_^20^ = −22.3° (c = 1.0, CH_3_OH). Selected NMR signals are presented in Table 3. HRMS (ESI): *m/z* calcd for C_15_H_20_O_10_H ([M + H]+) 361.11292; found 361.11278; calcd for C_15_H_20_O_10_Na ([M + Na]+) 383.09487; found 383.09494.

2′,5-Di-*O*-(3,4,5-tri-*O*-methylgalloyl)-α,β-D-hamamelofuranose (**9e**). Colourless waxy solid; α:β = 0.4:1; [α]_D_^20^ = +20.2° (c = 1.0, CH_3_OH). 7.42 (s, H-Arβ), 7.39 (s, H-Arβ), 7.26 (s, H-Arα), 7.25 (s, H-Arα), 5.32 (s, H-1α), 5.27 (s, H-1β), 4.60 (dd, *J* = 12.2, 2.7 Hz, 5aα), (d, 4.56–4.48 (m, H-5aβ, H-5bα), 4.54, *J* = 11.5, H-2′aβ), 4.50, *J* = 11.6, H-2′bβ), 4.43 (dd, *J* = 12.0, 4.4 Hz, H-5bβ), 4.38 (d, *J* = 11.5 Hz, H-2′aα), 4.35 (d, *J* = 7.9 Hz, H-3β), 4.32–4.27 (m, H-4α), 4.32 (d, *J* = 11.5 Hz, 2′bα), 4.24 (ddd, *J* = 7.5, 4.3, 2.9 Hz, H-4β), 4.04 (d, *J* = 7.7 Hz, H-3α), 3.86 (s, 2 × OCH_3_), 3.85 (s, 2 × OCH_3_), 3.82 (s, 2 × OCH_3_), 3.81 (s, 4 × OCH_3_), 3.80 (s, OCH_3_α), 3.79 (s, OCH_3_β). ^13^C NMR (101 MHz, CD_3_OD) δ 167.7 (COOβ), 167.6 (COOβ), 167.3 (COOα), 167.1 (COOα), 154.3 (4 × C-OCH_3_β), 154.3 (4 × C-OCH_3_α), 143.7 (C-OCH_3_α), 143.6 (C-OCH_3_α), 143.6 (C-OCH_3_β), 143.5 (C-OCH_3_β), 126.6 (C-Arβ), 126.4 (C-Arβ), 126.1 (C-Arα), 126.0 (C-Arα), 108.3 (CH-Arβ), 108.2 (CH-Arβ), 108.1 (CH-Arα), 107.9 (CH-Arα), 102.9 (C-1β), 99.1 (C-1α), 81.5 (C-4β), 80.8 (C-4α), 80.7 (C-2β), 77.8 (C-2α), 73.5 (C-3β), 72.5 (C-3α), 67.9 (C-2′β), 67.0 (C-2′α), 65.9 (C-5β), 65.3 (C-5α), 61.1 (2 × OCH_3_β, 2 × OCH_3_α), 56.7 (4 × OCH_3_β), 56.6 (4 × OCH_3_α). HRMS (ESI): *m/z* calcd for C_26_H_32_O_14_Na ([M + Na]^+^) 591.16843; found 591.16848.

Mono-*O*-(3,4,5-tri-*O*-methylgalloyl)-α,β-D-hamamelofuranose (**10e**). Colourless waxy solid; [α]_D_^20^ = −8.8° (c = 1.0, CH_3_OH). Selected NMR signals are presented in Table 3. HRMS (ESI): *m/z* calcd for C_16_H_22_O_10_Na ([M + Na]^+^) 397.11052; found 397.11090.

2′,3,5-Tri-*O*-(3,4,5-tri-*O*-methylgalloyl)-α,β-D-hamamelofuranose (**11e**) and 1,2′,5-tri-*O*-(3,4,5-tri-*O*-methylgalloyl)-α,β-D-hamamelofuranose (**12e**). Colorless waxy solid; **12e**α:**11e**α:**12e**β:**11e**β = 0.1:0.4:0.8:1; Selected signals ^1^H NMR (400 MHz, CD_3_OD) δ 6.54 (s, H-1α-**12e**), 6.31 (s, 1H, H-1β-**12e**), 5.40 (s, H-1α-**11e**), 5.32 (s, H-1β-**11e**), ^13^C NMR (101 MHz, CD_3_OD) δ 103.8 (C-1β-**11**), 98.9 (C-1α-**11**), 101.8 (C-1β-**12**). HRMS (ESI): *m/z* calcd for C_36_H_42_O_18_Na ([M + Na]^+^) 785.22634; found 785.22703.

2′,5-Di-*O*-(3,4,5-tri-*O*-benzylgalloyl)-α,β-D-hamamelofuranose (**9g**). White solid; α:β = 1:0.6; [α]_D_^20^ = +41.9° (c = 1.0, CHCl_3_). ^1^H NMR (400 MHz, CDCl_3_+CD_3_OD, 40 °C) δ: 7.49, 7.41, 7.38, 7.34 (4 × s, 2 × H-Arβ, 2 × H-Arα), 7.45–7.30 (m, 48 × CH-Ph), 7.29–7.21 (12 × m, CH-Ph), 5.34 (s, H-1α), 5.28 (s, H-1β), 5.14–5.03 (m, 12 × CH_2_), 4.66 (dd, *J* = 12.2, 2.8 Hz, 5aα), 4.61–4.47 (m, H-5aβ, 2′aβ, 2′bβ, H-5bβ), 4.43 (dd, *J* = 12.3, 5.7 Hz, H-5bα),4.34 (s, H-2′aα,H-2′bβ), 4.29 (d, *J* = 7.9 Hz, H-3β), 4.29–4.23 (m, H-4α, H-4β), 3.89 (d, *J* = 8.1 Hz, H-3α). ^13^C NMR (101 MHz, CDCl_3_+CD_3_OD, 40 °C) δ 166.7 (COOβ), 166.5 (COOβ), 166.2 (COOα), 166.0 (COOα), 152.5, 152.4 (4 × C-OCH_2_α, 4 × C-OCH_2_β), 142.6 (C-OCH_2_α), 142.5 (C-OCH_2_α), 142.4 (C-OCH_2_β), 142.3 (C-OCH_2_β), 137.2 (8 × C-Ph), 136.5 (8 × C-Ph), 128.5, 128.5, 128.4, 128.4 (24 × CH-Ph) 128.1, 128.1 (8 × CH-Ph), 128.0, 127.9, 127.9, 127.9 (12 × CH-Ph), 127.5, 127.5 (16 × CH-Ph), 124.9 (C-Arβ), 124.8 (C-Arβ), 124.7 (C-Arα), 124.3 (C-Arα), 109.2 (CH-Arβ), 109.1 (CH-Arβ, CH-Arα), 109.0 (CH-Arα), 101.5 (C-1β), 97.4 (C-1α), 80.7 (C-4β), 79.4 (C-2β), 79.1 (C-4α), 77.3 (C-2α), 75.1 (4 × CH_2_-Ph), 72.7 (C-3β), 71.7 (C-3α), 3 × 71.1, 71.0 (8 × CH_2_-Ph), 66.7 (C-2′β), 66.1 (C-2′α), 65.5 (C-5β), 64.3 (C-5α). HRMS (ESI): *m/z* calcd for C_62_H_56_O_14_H ([M + H]^+^) 1025.37428; found 1025.37038; calcd for C_62_H_56_O_14_Na ([M + Na]+) 1047.35678; found 1047.35623.

Mixture of mono-*O*-(3,4,5-Tri-*O*-benzylgalloyl)-α,β-D-hamamelofuranoses (**10g**). White solid; [α]_D_^20^ = −10.9° (c = 1.0, CH_3_OH). Selected NMR signals are presented in Table 3. HRMS (ESI): *m/z* calcd for C_34_H_34_O_10_Na ([M + Na]^+^) 625.20442; found 625.20470.

### 2.4. Removal of Protecting Groups

#### 2.4.1. Simultaneous Deacetylation and Deisopropylidenation of **6a** and **7**

Diacylated compound **6a** (0.1 mmol) or triacylated compound **7** (0.1 mmol) were dissolved in CH_3_CN (2 mL), and 3M HCl (2 mL) was added. The reaction mixture was stirred at laboratory temperature for 72 h. After the reaction, the organic solvent was removed under reduced pressure and the aqueous residue was extracted with ethyl acetate (3 × 50 mL). The organic layer was washed with brine (3 × 50 mL), dried (Na_2_SO_4_), and concentrated under reduced pressure. The product **1** from the deprotection of digallate **6a** was obtained as a pure compound in high yield (94%), while the reaction mixture obtained from the deprotection of compound **7** contained the product **1** and gallic acid and after chromatography on silica gel (0.1%CH_3_COOH in EtOAc) only 56% of product **1** was obtained.

#### 2.4.2. Debenzylation of **9g**

Diacylated hamamelofuranose **9g** (0.206 g, 0.02 mmol) was dissolved in MeOH (10 mL) and 10% Pd/C (0.06 g) was added. The reaction mixture was intensively stirred at room temperature (25 °C) under hydrogen atmosphere for 18 h and then filtered through Celite 545. After washing the celite cake with MeOH, the filtrate was concentrated to give 0.094 g (97%) of product **1**.

#### 2.4.3. Hamamelitannin (**1**)

Amorphous white solid; α:β = 0.7:1; [α]_D_^20^ = +37.9° (c = 1.0, CH_3_OH). ^1^H NMR (400 MHz, CD_3_OD) δ: 7.12 (s, H-Arβ), 7.12 (s, H-Arβ), 7.10 (s, H-Arα), 7.08 (s, H-Arα), 5.34 (s, H-1α), 5.26 (s, H-1β), 4.53 (dd, *J* = 12.1, 2.6 Hz, H-5aα), 4.52 (dd, *J* = 11.7, 2.9 Hz, 5aβ), 4.44 (s, H-2′aβ, H-2′bβ), 4.35–4.24 (m, H-5bα, H-5bβ, H-2′aα, H-2′bα, H-4α), 4.20 (ddd, *J* = 7.8, 4.8, 2.9 Hz, H-4β), 4.17 (d, *J* = 7.7 Hz, H-3β), 3.90 (d, *J* = 7.8 Hz, H-3α). ^13^C NMR (101 MHz, CD_3_OD) δ 168.6 (COOβ), 168.4 (COOβ), 168.2 (COOα), 168.2 (COOα), 146.5, 146.5, 2 × 146.4 (2 × C-OHβ, 2 × C-OHα), 140.0 (C-OHα), 139.9 (C-OHα), 139.8 (C-OHβ), 139.8 (C-OHβ), 121.5 (C-Arβ), 121.4 (C-Arβ), 121.2 (C-Arα), 121.0 (C-Arα), 110.3, 2 × 110.3, 110.2 (2 × CH-Arβ, 2 × CH-Arα), 102.9 (C-1β), 99.0 (C-1α), 81.4 (C-4β), 80.8 (C-2β), 80.3 (C-4α), 77.7 (C-2α), 74.2 (C-3β), 73.2 (C-3α), 67.4 (C-2′β), 67.1 (C-5β), 66.6 (C-2′α), 65.7 (C-5α). HRMS (ESI): *m/z* calcd for C_20_H_20_O_14_Na ([M + Na]^+^) 507,07508; found 507,07464.

## 3. Results and Discussion

### 3.1. Synthesis of Hamamelitannin by Acylation of Acceptor **2** Prepared from D-Ribose

D-Ribose was used as the starting material for the preparation of isopropylidenated D-hamamelofuranose **2** (Figure 1). Treatment of D-ribose with anhydrous acetone in the presence of a catalytic amount of concentrated sulfuric acid yielded the corresponding 2,3-acetonide **2** [46] in 91% yield. The reaction of **2** with aqueous formaldehyde in the presence of potassium carbonate in MeOH introduced the branching hydroxymethyl group through a Ho crossed aldol reaction [47].

In the initial stage of the work, an acetyl-protecting group was selected for the galloyl moiety because the deacetylation of phenols and the removal of the isopropylidene group from the sugar can be performed in one step under acidic conditions. In the first step, standards of the desired triacetylgallates of hamamelofuranose **8** were chemically prepared. After the preparation of 3,4,5-tri-*O*-acetylgalloyl chloride (**3**) from gallic acid in two steps, we looked for optimal conditions for the acylation of hamamelose **2**. Acetylated phenols are sensitive to both acidic and basic conditions, but deprotected gallates can oligomerize under basic conditions. Therefore, in the synthesis, it was necessary to find slightly basic or neutral conditions that allow 2′,5-di-*O*-acylation of compound **2** and to which the acetyl groups are inert. Our secondary goal was to regioselectively achieve 2′,5-di-*O*-acylated hamamelose **6a** in maximum yields, using a minimum of equivalents of acyl reagent **3**. A high content of monoacylated or triacylated products was undesirable.

Several conventional acylation methods have been used, operating from mildly basic to neutral conditions. The results are summarized in Table 1. In initial experiments, we investigated mild basic reaction conditions working with acetylated galloyl chloride **3**, which has been reliably verified in many acylation reactions [53,54,55]. A mixture of two bases—Et_3_N and DMAP (1 equiv. and 0.25 equiv. relative to 1 equiv. of acyl) was used in dichloromethane (Table 1, Entry 1, 2). Using 2.2 equivs of acyl reagent **3** and appropriate equivalents of bases led to the mixture of per-*O*-acetylated trigallate **7**, digallate **6a**, and relatively high content of monogallates (24%), while conditions for theoretical attachment of three acyl groups (3.3 equivs of **3**) led to **7** as a major product (81%). Digallate **6a** and monogallates were formed in minimal quantities.

As a very effective tool for providing regioselective acylations of sugars, procedures using organotin reagent–dibutyltin oxide (Bu_2_SnO) [56,57]. Treatment of **2** by acyl donor **3** (2.2 equivs) and Bu_2_SnO (2.2 equivs) provided digallate **6a** in 84% yields and only minor quantities of trigallate and monogallates were isolated (Table 1, Entry 5). Another modification of the method (Bu_2_SnO quantity, solvent changed to CH_3_CN) led to an increase in the amount of monogallates (Table 1, Entry 3, 4).

In one of our previous investigations, we studied ZnO as a convenient catalyst in the 4-*O*-acetylferuloylation of glycosides [58]. Therefore, we have examined tri-*O*-acetylgalloylation of **2** in CH_2_Cl_2_ (Table 1, Method C, Entry 6), but we isolated a mixture of acetylated monogallates as the main product fraction. Better results were obtained when we used CH_3_CN as the reaction medium (Table 1, Entry 7, 8). With elevating the reaction temperature to 40 °C, ZnO equivalents and reaction time lead to trigallate **7** a digallate **6a** in the summary yield 83% (Table 1, Entry 8). It is interesting that we did not observe compound **7** as the β-anomer and **7α** was the exclusive tri-*O*-acylated product under all conditions; this suggests the neighboring group effect by the C-2′ acyloxymethyl group in the rigid 2,3-isopropylidenated furanose ring, as has been already reported [59].

Recently, we have been intensively dealing with enzymatic acylations of sugars by various phenolic acid donors [43,48,60,61]. The commercial lipase Lipozyme TL IM was shown to be the most effective catalyst in terms of its substrate specificity, reactivity, and stability in these reactions. The disadvantage of its use was the longer reaction time, reaching several days. We have, therefore, galloylated **2** under our optimised conditions for galloylation of methyl β-D-glucopyranoside [61] catalysed by Lipozyme TL IM using 3 equivalents of vinyl gallate **4b** as an acyl donor at 37 °C and dry *t*-butyl alcohol (*t*-BuOH) as a solvent (Figure 2). The 2,3-isopropyl-D-hamamelofuranose **2** was surprisingly better accepted by the enzyme than methyl β-D-glucopyranoside. The reaction proceeded regioselectively to the 5-OH position of compound **2** and the maximum yield (66%) of monogallate **5b** was obtained after 242 h (Figure 2). *t*-BuOH is quite a polar solvent (octanol/water partition coefficient as log Pow: 0.30). According to our experience, the lipase-catalyzed transesterification proceed faster in less polar solvents such as for example *t*-butyl methyl ether (*t*-BuMeO) (octanol/water partition coefficient as log Pow: 1.06).

The same reaction of **2** with 3 equiv. of **4b** was repeated in *t*-BuMeO. Under these conditions, after 61 h it gave 5-*O*-gallate **5b** in a yield of 82%. The reaction was significantly faster with a higher yield and the product was again the monoacylated product **5b** (Figure 2). Regarding the increase in reaction rate, we can hypothesize that *t*-BuMeO opens the hydrophobic lid in the active center of Lipozyme TL IM more efficiently than *t*-BuOH, and the substrate binding site is more accessible [62].

The lipase specificity for the acyl donor structure can significantly influence the course of the reaction and the degree of acylation, as has been demonstrated for another commercial lipase from *Thermomyces lanuginosus*—Lipolase 100T [48]. During acylation of α-glucopyranoside with phenolic vinyl esters in CH_3_CN, Lipolase catalyzed the formation of only 6-*O*-acylated products. If non-phenolic vinyl esters were used, 2,6-di-*O*-acylated glucopyranoside was also formed. Therefore, we have used acetylated vinyl gallate **4a** (3 equiv.) in the same transesterification reactions catalyzed by Lipozyme TL IM in *t*-BuMeO, i.e., the phenolic groups were hydrophobized by acetyls. (Figure 1). After 19 h, the starting compound **2** was consumed and more products (also UV inactive) were visible on the TLC plate. After column chromatography on silica gel, three main UV active products were obtained: 5-monoacylated product **5a** (9%), 2′,5-diacyl **6a** (6%), and 34% of an unexpected product—2,3-*O*-isopropylidene-2′-*O*-acetyl-5-*O*-(3,4,5-tri-*O*-acetylgalloyl)-α,β-D-hamamelofuranose (**5a**-2′-Ac) (Figure 1, Method D). Lipozyme can probably use the triacetylated vinyl gallate as an activated acetyl donor, and the remaining UV inactive products were partially acetylated derivatives of **2**.

### 3.2. Synthesis of Hamamelitannin by Enzymatic Acylation of D-Hamamelose

Promising results with enzymatic galloylation of compound **2** prompted our increased efforts to prepare hamamelitannin **1** via direct enzymatic acylation of D-hamamelose (**8**). D-hamamelose, although a rare branched sugar, is commercially available. One of the ways to prepare **8** is molybdic acid-catalyzed isomerization of D-fructose by Bílik reaction [63,64].

At first, we investigated the ability of Lipozyme to acylate **8** with a routine commercial aromatic donor—vinyl benzoate (**4c**). We were inspired by work [65] in which D-fructose benzoylated with vinyl benzoate (3 equiv.) using lipase from *C. antarctica* B (CAL) and lipase from *Mucor miehei* (MML) in *t*-BuMeO gave after 7 h 1,6-di-*O*-benzoyl-D-fructofuranoside (80%). In our hands, using 4 equiv. of benzoate **4c**, a Lipozyme-catalyzed reaction in *t*-BuMeO for 50 h afforded di-*O*-benzoate **9c** (37%), traces of monobenzoates **10c** and a diverse mixture of tri-*O*-benzoates **11c**, **12c**, and **13** in 55% total yield (Figure 3). The high proportion of variously benzoylated secondary hydroxyl groups, even the existence of tribenzoate **13** in pyranose form, indicated that the reaction with reactive hydrophobic aromatic donors would not be selective.

When proceeding the reaction with 3 equivalents of acetylated vinyl gallate **4a** and gallate **4b** under similar conditions (Lipozyme TL IM, *t*-BuMeO, 37 °C), the desired products were obtained in both cases, however, in very low yields. Acetylated **4a** gave 13% of diacyl **9a** after 41 h and the rest were various UV-inactive acetylated products. The reaction with vinyl gallate **4b** was allowed to react for a longer time (292 h), and again obtained only 15% of the acylation product **1** (Figure 4). The monoacylated product’s content was not visible on TLC. In both cases, we did not observe the presence of aromatic triacylated products. Vinyl gallate **4b** was not sufficiently reactive in *t*-BuOH and no galloylation product with hamamelose **8** was observed even after 10 days.

The structure of the starting compounds in lipase-catalyzed esterifications or transesterifications, especially in the case of phenolic compounds, influences the course of the reaction [48,66]. Therefore, we decided to test other more hydrophobic and stable vinyl esters of gallic acid derivatives in the investigated enzyme reaction. Vinyl esters of syringic acid (**4d**) and 3,4,5-trimethoxybenzoic acid (**4e**) were prepared according to our previous work [48]. Two new vinyl esters were also prepared—3,4,5-tri-*O*-(*t*-butyldimethylsilyl)gallate (**4f**) and 3,4,5-tri-*O*-benzylgallate (**4g**). The silyl derivative **4f** was prepared by silylation of vinyl gallate and the benzylated derivative **4g** was prepared by transesterification of 3,4,5-tribenzylgallic acid with vinyl acetate. Silyl and benzyl protective groups are widely used in the syntheses, as they can be effectively removed under mild, relatively neutral conditions [67].

The studied enzymatic acylations were performed according to previously implemented conditions. (1 mmol of **8**, 3 equiv. of vinyl ester, 0.2 g of Lipozyme TL IM, 20 mL of solvent, 37 °C). To compare the effect of solvent on the reaction time, the composition of products. and product yields with individual acyl donors; these were carried out in *t*-BuOH as well as in *t*-BuMeO. The reaction with the benzylated gallate **4g** was also carried out in CH_3_CN. Acylations were monitored by TLC chromatography. The reactions were stopped when the concentration of the products no longer increased. The lipase and molecular sieves were filtered off and the filtrate was purified by chromatography after concentration.

The results of the reaction (Figure 5) summarized in Table 2 showed that Lipozyme TL IM catalyzes acylations with all acyl donors except silylated gallate **4f**. In the case of **4f**, we did not observe any product in both solvents even after hundreds of hours (Table 2, entries 5, 6). Acyl donor **4f** is probably too large to interact with the active site of the enzyme. Acylation with syringate **4d** proceeded as within the longest reaction times (more than 200 h), while hydrophobic **4e** and **4g** reacted faster (tens of hours). This is consistent with our previous experience [48], and it appears that the transesterification activity of Lipozyme TL IM, similarly to Lipolase 100T (both are lipases from *Thermomyces lanuginosus*), corresponds to the hydrolytic activity of type A feruloylesterase [68]. In general, reactions in *t*-BuOH proceeded slower and a higher quantity of monoacylated products were isolated (Table 2, entries 1, 3, 7). On the contrary, we have observed only negligible amounts of monoacyl-hamameloses in *t*-BuMeO and mostly 2′,5-di-*O*-acyls of α,β-D-hamamelofuranose (**9d–e**, **9g**) (Figure 5) were isolated (Table 2, entries 2, 4, 9). The reactivity of trimethoxybenzoate **4e** was similar to that observed for benzoate **4c**. Products with acylated secondary hydroxyls were also observed. We isolated a significant proportion of triacyls in *t*-BuMeO for **4e** (Table 2, entry 4), and the highest yields of diacyls for *t*-BuOH were achieved using **4e** (Table 2, entry 3). This suggests that the reaction was directed towards the products that were more soluble in the used solvent.

The acylation of **8** with the bulky benzylated acyl donor **4g** had a different reaction course. The reaction in *t*-BuMeO proceeded with the highest yield (84%) of 2′,5-diacyl **9g** (Table 2, entry 9). The desired main product **9g** was the only one precipitated from the reaction mixture. After the end of the reaction, it was filtered together with the immobilized enzyme and molecular sieves. It was then washed with hot ethyl acetate from the filter cake. A similar reaction system, in which the starting monosaccharide (D-hamamelose), biocatalyst and the product (**9g**) were insoluble or almost insoluble in the reaction solvent, which serves as an adjuvant, were known from enzymatic syntheses of sugar fatty acids esters [69]. The low solubility of **9g** in the reaction medium probably protects it from unwanted acylations to the secondary hydroxyls. Reactions in more polar solvents (*t*-BuOH, CH_3_CN) proceeded more slowly with low product conversion. This could indicate that the lipase does not have a favorable conformation for the bulky acyl **4g** or is inactivated by the products.

We also examined the structural composition of our mixtures of monoacylated products isolated from reactions in *t*-BuOH. Theoretical structures in the mixture of hamamelose monoacyls are shown in Figure 6. They could not be separated individually, but their mixtures were analyzed by NMR, and the H-1 and C-1 signals for individual anomers and conformations were assigned with the support of literature data. The available literature and experimental values are listed in Table 3. These data demonstrate that the initial acylation is not selective (Entries 9–14). In the mixture, 5-*O*-monoacylated and predominating 2′-*O*-acylated hamameloses are visible. Generally, the primary 2′-OH position is sterically less favorable than the primary 5-OH position. It is possible that the acylation takes place first in the 2′-OH position if hamamelose is present in the reaction medium in pyranose form. This acylated pyranose is then transformed into furanose via mutarotation.

### 3.3. Deprotections for Obtaining Hamamelitannin ***1***

In the final step, the tri-*O*-acetylgalloylated compounds **6a**, **7** and benzylgalloylated **9g** were deprotected. Acidic conditions—3M HCl in CH_3_CN—were found to be sufficient for simultaneous deisopropylidenation and deacetylation, while the galloyl groups were retained. The product **1** from the deprotection of diacyl **6a** was obtained as a pure compound in high yield (94%, conditions (a) in Figure 7), while the reaction mixture obtained from the deprotection of compound **7** contained **1** and gallic acid. Gallic acid originated from deacylation of anomeric gallate moiety sensitive to acidic conditions. The disadvantage of this method was the long reaction time (3 days at laboratory temperature). Debenzylation of **9g** by reductive cleavage with molecular hydrogen over 10% Pd/C proceeded smoothly. After 18 h, the reaction mixture was filtered through Celite 545, and after the concentration of the filtrate, hamamelitannin **1** was obtained with a 97% yield and satisfactory purity (conditions (b), Figure 3).

## 4. Conclusions

2,3-Isopropylhamamelofuranose **2** and D-hamamelose **8** were studied as acceptors in chemoenzymatic galloylations with the aim of developing an efficient preparation of hamamelitannin. The chemical preparation of hamamelitannin from furanose **2** proceeded smoothly. Base-catalyzed acylation of **2** with acetylated galloyl chloride **3** provided 81% of 1,2′,5-trigallate **7**. The Bu_2_SnO-promoted reaction yielded 84% of 2′,5-digallate **6a** regioselectively. Enzymatic reactions using vinyl gallate **4b** or its acetylated analogue **4a** catalyzed by Lipozyme TL IM provided mainly 5-*O*-galloyl derivatives. Reaction condition using acyl donor **4b** in *t*-BuMeO afforded 82% 5-*O*-gallate **5b** after 61 h. The pilot enzymatic benzoylation of hamamelose **8** using vinyl benzoate and Lipozyme TL IM as a biocatalyst gave mainly benzoylated furanoses (89%), of which mainly tribenzoates (52%). Similar reactions with vinyl gallate **4b** and its acetylated analogue **4a** gave 2′,5-diacylated hamameloses but in yields below 20%. Acetylated vinyl gallate **4a** also appeared as an acetyl donor. The Lipozyme TL IM, in its presence in the reaction mixture, also performed acetylation of acceptors **2** and **8**. Hamamelose **8** in *t*-BuMeO with vinyl gallates, where phenolic groups were hydrophobized with methyl or benzyl moiety, readily afforded 2′,5-diacylated hamamelofuranoses (65–84%), with the exception of the reaction with the silylated gallate **4f**. The best results were obtained with tribenzylated gallate **4g**, where the desired 2′,5-diacyl **9g** precipitated from the reaction mixture. Similar reactions in more polar solvent *t*-BuOH gave mainly monoacyls and proceeded more slowly. They did not proceed on secondary hydroxyls and were not regioselective on primary hydroxyls. Finally, after deacetylation and deisopropylidenation of compound **6a** under acidic conditions, product **1** was obtained in 94% yield (79% after two steps). Similarly, after the reductive debenzylation of compound **9g**, hamamelitannin **1** was obtained with a yield of 97% (82% after two steps from hamamelose). The accomplished syntheses (especially the enzymatic method) open the way to multigram preparations of bioactive hamamelitannin and its analogs.

## Data Availability

Data are contained within the article.

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
