# Peer review of "From Hamamelitannin Synthesis to the Study of Enzymatic Acylations of D-Hamamelose"

_biomolecules, 2023, doi:10.3390/biom13030519_

Round 1
Reviewer 1 Report
Authors:
The manuscript on enzymatic acylations of D-hamamelose represents an important approach in the field of investigation of enzymatic processes. The manuscript is easy to read, well organized and logical. The experiments were well planned and are well documented.
I have found relatively high number of technical insufficiencies. I have marked only some of them by color (NOT all of them!) in the downloaded manuscript. Please, find it attached herewith.
I recommend to correct ALL mistakes and errors, and re-submit the corrected manuscript.

Author Response
Comments and Suggestions for Authors
Authors:
The manuscript on enzymatic acylations of D-hamamelose represents an important approach in the field of investigation of enzymatic processes. The manuscript is easy to read, well organized and logical. The experiments were well planned and are well documented.
Thank you.
I have found relatively high number of technical insufficiencies. I have marked only some of them by color (NOT all of them!) in the downloaded manuscript. Please, find it attached herewith.
I recommend to correct ALL mistakes and errors, and re-submit the corrected manuscript.
Thank you for alerting us to these minor technical errors that arose from inattention and time constraints. We went through the manuscript and hopefully have removed all errors and typos.
Reviewer 2 Report
The submitted manuscript reported efficient preparation of hamamelitannin through chemical and enzymatic ways. This work filled a long-standing gap in the total synthesis of hamamelitannin. Overall, this is an interesting and valuable studying. However, as indicated below, there are some minor flaws remaining and should be addressed by the authors to improve the quality of this work.
Minor Critiques:
1. The article's introduction provides a well-elaborated explanation of the role and efficacy of hamamelitannin. To provide readers with a clearer understanding of ongoing research and production of the substance, it is suggested to list several more literatures on the preparation of hamamelitannin. Is it commercially available? how is it produced?
2. Since the production of hamamelitannin was quite rare, they can probably introduce the production of other tannin substance. Tannin substance shared similar structure and bioactivities.
3. The name of the country where the chemical supplier is located should be accompanied by the company name.
4. The units in the materials and methods section should be standardized to 'h' or 'hours'. Note that there should be no space between the temperature unit and the numbers.
5. Conclusions should summarize the significant findings, and thus needs to be extended further within the scopes.
Author Response
Comments and Suggestions for Authors
The submitted manuscript reported efficient preparation of hamamelitannin through chemical and enzymatic ways. This work filled a long-standing gap in the total synthesis of hamamelitannin. Overall, this is an interesting and valuable studying. However, as indicated below, there are some minor flaws remaining and should be addressed by the authors to improve the quality of this work.
We appreciate notices of minor flaws in the article and the help to improve the quality of this manuscript.
Minor Critiques:
- The article's introduction provides a well-elaborated explanation of the role and efficacy of hamamelitannin. To provide readers with a clearer understanding of ongoing research and production of the substance, it is suggested to list several more literatures on the preparation of hamamelitannin. Is it commercially available? how is it produced?
Yes, hamamelitannin is commercially available and quite expensive. It is mostly isolated from witch hazel and sweet chestnut for commercial use, as declared for example by Carl Roth, Biosynth, BOC Sciences, Phytolab or BioCrick. I have included information in this sense in the introduction.
- Since the production of hamamelitannin was quite rare, they can probably introduce the production of other tannin substance. Tannin substance shared similar structure and bioactivities.
In the Introduction, I put hamamelitannin in the group of tannins, more precisely hydrolyzable tannins. I added six references on the classification, structure, properties, and synthesis of relevant tannins.
- The name of the country where the chemical supplier is located should be accompanied by the company name.
We added the city and the name of the country where the chemical supplier is located behind the company name in Section 2.1.
- The units in the materials and methods section should be standardized to 'h' or 'hours'. Note that there should be no space between the temperature unit and the numbers.
We standardized time units in the materials and methods section to "h". As for the temperature, we are confused. We looked at the articles in Biomolecules and other journals and everywhere there is a space between the number and the temperature unit (e.g. 80 °C). We will probably get advice from the technical editor when preparing the proof. But in Chapter 2.2.1 we adjusted 80 ° C to 80 °C. Maybe this is what you mean.
- Conclusions should summarize the significant findings, and thus needs to be extended further within the scopes.
The conclusions were largely expanded.
Reviewer 3 Report
The Authors describe the total synthesis of hamamelitanin. This is the compound which is easily accessible from natural sources and already widely used in cosmetics. Hence my hesitation is about the sense of the attempts to perform its total syntheses. On the other hand, the paper is very well written and provides all necessary experimental details. Although the methods used, particularly the enzyme promoted acylation of alcohols, are known, their successful application to the new and demanding substrates should be emphasized. Therefore, in spite of my reservations, I am of the opinion that the paper deserves publication in the present form
Author Response
Comments and Suggestions for Authors
The Authors describe the total synthesis of hamamelitanin. This is the compound which is easily accessible from natural sources and already widely used in cosmetics. Hence my hesitation is about the sense of the attempts to perform its total syntheses. On the other hand, the paper is very well written and provides all necessary experimental details. Although the methods used, particularly the enzyme promoted acylation of alcohols, are known, their successful application to the new and demanding substrates should be emphasized. Therefore, in spite of my reservations, I am of the opinion that the paper deserves publication in the present form
We are grateful for your opinion.
Reviewer 4 Report
Reviewer’s Comments:
The manuscript “From Hamamelitannin Synthesis to the Study of Enzymatic Acylations of D-Hamamelose” is very interesting work. The bioactive natural substance, hamamelitanin, was effectively synthesized in two ways. Chemical acylation of 2,3-O-isopropylidene-α,β-D-hamamelofuranose with 3,4,5-tri-O-acetylgalloyl chloride followed by deprotection provided hamamelitanin in 79%. Enzymatic galloylation of 2,3-O-isopropylidene-α,β-D-hamamelofuranose with vinyl gallate under catalysis of Lipozyme TL IM in t-butyl alcohol (t-BuOH) or t-butyl methyl ether (t-BuMeO) provided solely the 5-O-galloylated product. The reaction in t-BuMeO proceeded in shorter reaction time (61h) and higher yield (82 %). The more hydrophobic vinyl 3,4,5-tri-O-acetylgallate in the same reactions gave large amounts of acetylated products. However, the following issues should be carefully treated before publication.
1. In abstract, the author should add more scientific findings.
2. Keywords: the synthesized system is missing in the keywords. So, modify the keywords.
3. In the introduction part, the introduction part is not well organized and cited references should cite recently published articles such as 10.3390/molecules27196457, 10.3389/fchem.2022.1023316
4. Introduction part is not impressive and systematic. In the introduction part, the authors should elaborate the scientific issues in the Enzymatic Acylations research.
5. Results…, The author should provide reason about this statement “Silyl and benzyl protective groups are widely used in the syntheses, as they can be effectively removed under mild, relatively neutral conditions”.
6. The authors should explain regarding the recent literature why “The acylation of 8 with the bulky benzylated acyl donor 4g had an interesting course”.
7. The author should explain the latest literature “We also looked at the structural composition of a mixture of monoacylated products isolated from reactions in t-BuOH”.
9. Comparison of the present results with other similar findings in the literature should be discussed in more detail. This is necessary in order to place this work together with other work in the field and to give more credibility to the present results.
10. The conclusion part is very week. Improve by adding the results of your studies.
Author Response
Reviewer’s Comments:
The manuscript “From Hamamelitannin Synthesis to the Study of Enzymatic Acylations of D-Hamamelose” is very interesting work. The bioactive natural substance, hamamelitanin, was effectively synthesized in two ways. Chemical acylation of 2,3-O-isopropylidene-α,β-D-hamamelofuranose with 3,4,5-tri-O-acetylgalloyl chloride followed by deprotection provided hamamelitanin in 79%. Enzymatic galloylation of 2,3-O-isopropylidene-α,β-D-hamamelofuranose with vinyl gallate under catalysis of Lipozyme TL IM in t-butyl alcohol (t-BuOH) or t-butyl methyl ether (t-BuMeO) provided solely the 5-O-galloylated product. The reaction in t-BuMeO proceeded in shorter reaction time (61h) and higher yield (82 %). The more hydrophobic vinyl 3,4,5-tri-O-acetylgallate in the same reactions gave large amounts of acetylated products. However, the following issues should be carefully treated before publication.
- In abstract, the author should add more scientific findings.
The abstract has been revised and additional details have been added. This however means that we exceed the limit of 200 words.
- Keywords: the synthesized system is missing in the keywords. So, modify the keywords.
Keywords have been modified. We focus more on the enzymatic acylation. The chemical method was mainly used to prepare the standard.
- In the introduction part, the introduction part is not well organized and cited references should cite recently published articles such as 10.3390/molecules27196457, 10.3389/fchem.2022.1023316
Topics of the papers proposed to cite, i.e. „the synthesis of 1,3,4-thiadiazole-fused-[1,2,4]-thiadiazole incorporating 1,4-benzodioxine moiety“ and „benzimidazole-based thiazole derivatives as multipotent inhibitors of α-amylase and α-glucosidase“, both from authors Hussain et al., are not related to our article at all and we will not cite them. The recently published articles on the topic of hamamelitannin are already included in the list of references.
- Introduction part is not impressive and systematic. In the introduction part, the authors should elaborate the scientific issues in the Enzymatic Acylations research.
We have added a paragraph to the Introduction that provides general information on enzyme acylations and we have supplemented the relevant literature.
- Results…, The author should provide reason about this statement “Silyl and benzyl protective groups are widely used in the syntheses, as they can be effectively removed under mild, relatively neutral conditions”.
We have added a suitable citation (Reference 65), where the relevant, generally known techniques in organic synthesis are mentioned, also with citations.
- The authors should explain regarding the recent literature why “The acylation of 8 with the bulky benzylated acyl donor 4g had an interesting course”.
The sentence was slightly changed, the word “interesting” was replaced by the word “different”. We think that the following sentence "The desired main product 9g was the only one precipitated from the reaction mixture" sufficiently explains the difference from other reactions. A similar reaction system in which the biocatalyst and the product are insoluble in the reaction solvent (which serves as an adjuvant) and known from the enzymatic synthesis of sugar fatty acids esters, is cited (Reference 67).
- The author should explain the latest literature “We also looked at the structural composition of a mixture of monoacylated products isolated from reactions in t-BuOH”.
For better clarity, the sentence was slightly changed: “We also examined the structural composition of our mixtures of monoacylated products isolated from reactions in t-BuOH.” We do not understand what the latest literature is supposed to explain. The sentence refers to our experiments.
- Comparison of the present results with other similar findings in the literature should be discussed in more detail. This is necessary in order to place this work together with other work in the field and to give more credibility to the present results.
This is a challenging task, as there are no published enzyme acylations to prepare hydrolyzable tannins, we have not found a reference that discusses enzyme galloylations in general, apart from our earlier work on glucopyranoside, which is cited. We compared the results of benzoylation on hamamelose with the results of benzoylation of fructose with other lipases (Reference 63).
- The conclusion part is very week. Improve by adding the results of your studies.
The conclusion part was expanded and provided with more details.